# DI-5-Cuffs: Lumbar Intervertebral Disc Proteoglycan and Water Content Changes in Humans after Five Days of Dry Immersion to Simulate Microgravity

**DOI:** 10.3390/ijms21113748

**Published:** 2020-05-26

**Authors:** Treffel Loïc, Navasiolava Nastassia, Karen Mkhitaryan, Jouan Emmanuelle, Zuj Kathryn, Gauquelin-Koch Guillemette, Custaud Marc-Antoine, Gharib Claude

**Affiliations:** 1Institut NeuroMyogène, Faculté de Médecine Lyon Est, 69008 Lyon, France; claude.gharib@univ-lyon1.fr; 2Centre de Recherche Clinique, Centre Hospitalier Universitaire d’Angers, 49100 Angers, France; Nastassia.Navasiolava@chu-angers.fr (N.N.); macustaud@chu-angers.fr (C.M.-A.); 3Siemens Healthinners, Service Application, 93210 Saint-Denis, France; karen.mkhitaryan@siemens-healthineers.com; 4Olea Medical, Service Application, 13600 La Ciotat, France; emmanuelle.jouan@olea-medical.com; 5Department of Kinesiology, University of Waterloo, Waterloo, ON N2L3G1, Canada; kathryn.zuj@gmail.com; 6CNES, Centre National d’Etudes Spatiales, 75001 Paris, France; guillemette.gauquelinkoch@cnes.fr; 7MitoVasc UMR INSERM 1083-CNRS 6015, Université d’Angers, 49100 Angers, France

**Keywords:** spine, intradiscal proteins, ADC diffusion, T1-T2 mapping, vertebral deconditioning, space physiology, back pain

## Abstract

Most astronauts experience back pain after spaceflight, primarily located in the lumbar region. Intervertebral disc herniations have been observed after real and simulated microgravity. Spinal deconditioning after exposure to microgravity has been described, but the underlying mechanisms are not well understood. The dry immersion (DI) model of microgravity was used with eighteen male volunteers. Half of the participants wore thigh cuffs as a potential countermeasure. The spinal changes and intervertebral disc (IVD) content changes were investigated using magnetic resonance imaging (MRI) analyses with T1-T2 mapping sequences. IVD water content was estimated by the apparent diffusion coefficient (ADC), with proteoglycan content measured using MRI T1-mapping sequences centered in the nucleus pulposus. The use of thigh cuffs had no effect on any of the spinal variables measured. There was significant spinal lengthening for all of the subjects. The ADC and IVD proteoglycan content both increased significantly with DI (7.34 ± 2.23% and 10.09 ± 1.39%, respectively; mean ± standard deviation), *p* < 0.05). The ADC changes suggest dynamic and rapid water diffusion inside IVDs, linked to gravitational unloading. Further investigation is needed to determine whether similar changes occur in the cervical IVDs. A better understanding of the mechanisms involved in spinal deconditioning with spaceflight would assist in the development of alternative countermeasures to prevent IVD herniation.

## 1. Introduction

Most astronauts experience back pain after spaceflight [1] and are more likely to develop disc herniations compared with a control population [2,3]. The process of disc herniation has been described in the literature; however, the underlying mechanisms of this pathology after spaceflight are not clearly defined [3,4]. Intervertebral disc content in space is not well described. Additionally, current exercise countermeasures used during spaceflight are not able to fully prevent spinal deconditioning [5]. A recent review article has highlighted the need for further research into the changes in spine health with spaceflight, making this area of investigation a priority for many space agencies [4]. The goal of the current study was to investigate intervertebral disc (IVD) pathophysiology to improve spine health after simulated long-duration spaceflight. 

Dry immersion (DI) is a ground-based model used to simulate the effects of the spaceflight environment on human physiology, particularly cardiovascular and musculoskeletal deconditioning [6,7] and postural disturbances [8]. DI has been commonly used in Russia since 1975 and in France since 2015. Thoraco-cephalic fluid shifts are particularly marked in this model [7,9], with hypovolemia responses comparable to spaceflight. Spinal deconditioning has been demonstrated in three-day dry immersion, with an increase in height and lumbar intervertebral disc swelling [10,11,12] shown by magnetic resonance spectroscopy analysis. However, the mechanisms of this change have not been investigated.

No previous studies have measured the proteoglycan content inside an IVD after simulated microgravity. Proteoglycans are a large component of IVDs, and have multiple forms and functions [13]. The largest domain of aggrecan contains the glycosaminoglycans-binding region. This protein domain is encoded by a single, very large (~4 kb) exon with ~120 Ser-Gly dipeptide repeats, which can generate >100 covalently-linked chondroitin sulfate chains. This collection of negatively-charged molecules constitutes the fixed charge density (FCD). Badlom et al. determined that IVD swelling during bed rest was proportional to the FCD, such that a 20% and a 45% reduction in the FCD resulted in a 25% and 55% reduction in disc water intake overnight, respectively [14]. Functionally, changes in diffusion have been highly correlated to IVD degeneration [15]. 

During real and simulated microgravity exposure, countermeasures, including resistance exercise programs, are used to counteract cardiovascular, muscle, bone, and vertebral deconditioning [16,17,18,19]. Leg or thigh cuffs have also been used during spaceflight as a method of mitigating the effects of thoraco-cephalic fluid shifts [19,20,21]. However, no work has investigated whether these cuffs also affect IVD properties. 

## 2. Objectives

The primary object of this work aimed to determine the mechanisms underlying IVD changes with simulated microgravity by analyzing IVD protein and fluid content. This study provided the opportunity to utilize a new, dynamic approach for assessing water motion and protein content with MRI diffusion analysis. Additionally, this study aimed to investigate the effects of thigh cuffs, used by Russian cosmonauts to ameliorate symptoms associated with cephalic fluid shift during spaceflight, on IVD properties.

## 3. Results

### 3.1. Plasma Volume

Overall, DI resulted in a 15–20% decrease in plasma volume (Table 1; *p* < 0.0001) with the application of thigh cuffs, resulting in a smaller plasma volume reduction (*p* = 0.09).

### 3.2. Blood Osmolality (mOsmol/kgH_2_0)

Blood sodium and potassium remained within the normal values at all of the measurements, and the blood osmolality remained unchanged. DI was accompanied by a significant increase in blood proteins in Control, but not in Cuffs after three days of DI (Table 2).

### 3.3. Spine Height and Lumbar Lordosis

There was a significant spinal lengthening induced by DI for all participants, with no difference in the degree of change between the two groups. On average, spine height increased from 59.81 to 61.06 cm (Figure 1). There was also a significant decrease (flattening) of the lumbar curvature after DI (Figure 2), with no difference between the two groups (−6 ± 12% on average).

### 3.4. MRI Diffusion: Lumbar Vertebral Disc Water Content

The ADC increased significantly in all of the lumbar discs, with an average increase of +7.34% ± 2.23%. There was no statistical difference in ADC between the Control and Cuffs groups (*p* > 0.05 NS); therefore, the results in Figure 3 are presented as the mean of all of the participants, with group values shown in Figure 4. No evidence of any pre-existing spinal pathologies were found before DI, and Pfirrmann grade remained constant following DI. 

### 3.5. MRI T1-Mapping: Protein Content Inside the Nucleus Pulposus

The relaxation time values in T1 increased in post DI (Figure 5). These results suggest a higher protein content (proteoglycans) inside IVD after simulated microgravity.

### 3.6. Back Pain

Subjects developed low to moderate back pain during immersion, especially during the first night (reported at the morning of D_2_). Back pain increased about three points (on a scale of 0 to 10) for the first two days, then decreased, without significant difference between groups (Figure 6).

## 4. Discussion

The current study was an investigation of spine physiology after exposure to five days of simulated microgravity utilizing dry immersion. The study provided the opportunity to investigate IVD water and protein changes in an effort to determine the mechanism of observed changes in spine health. The main findings of the study included the following: (a) significant spine lengthening with DI, (b) decreased lumbar lordosis, (c) increased ADC, indicating increased IVD water content, and (d) increased IVD protein content. Additionally, this study found no effects of a thigh cuff countermeasure on spine adaptations to DI. 

Spine lengthening is consistently observed during spaceflight [22,23,24], bed rest spaceflight simulation studies [11,25], dry immersion [6,10,26] and after a normal period of sleep [14,27]. The magnitude of change observed in the current study was consistent with results observed after only three days of DI [10]. Spine curvature was decreased with DI, but this could not fully explain the observed change in spine length. This suggests that when the spine is in weightlessness conditions, the IVD distance increases rapidly, with the annulus fibrosus fibers stretching to keep the IVD distance stable, despite longer exposure to simulated microgravity. To explain the observed IVD volume increase, two factors need to be considered [28]: spine loading caused by gravity, and the hydrostatic pressure of the extracellular fluid volume.

Spaceflight and simulated microgravity exposure present a condition of chronic unloading which decreases the lumbar lordosis and increases the risk of herniated discs [3,10,12,29,30,31]. Few studies have conducted IVD pressure measurements in vivo, with available data showing that pressure is dependent on body position and, potentially, time of day [27,32]. A study by Wilke et al. showed IVD pressures of 0.1 Mpa lying prone, 0.5 Mpa standing and 0.24 Mpa during sleep [27]. While measures of IVD pressure have not been conducted during real or simulated microgravity, the assumption can be made that the vertical force of gravity on the spine is reduced, leading to decreased IVD pressure and increased IVD volume.

It is well documented [33] that during lying or bed rest, plasma volume decreases are associated with a fluid and protein shift into interstitial areas. Plasma volume in the current study was estimated from measurements of hemoglobin and hematocrit. However, plasma volume changes are much smaller when estimated using protein variations [34]. This suggests a protein shift in the extracellular fluid volume, potentially affecting oncotic pressure and IVD protein and fluid content. However, the increased IVD oncotic pressure due to increased protein originating from plasma is likely very small.

Water and protein content were determined for all lumbar discs. Although all discs showed the same direction of change, variability existed in the magnitude of response. Body position during DI may partly explain these differences. During DI, participants typically rest with a degree of hip flexion, potentially resulting in a greater loss of curvature in the lower lumbar spine and the observed larger changes in the water and protein content.

The composition of extracellular fluid influences the IVD water content [33,35]. Proteoglycans, considered as water-traps [13], retain water and could explain why changes in the water and proteoglycan content are often linked. Johnstone et al. have demonstrated that under sustained compression, IVD water content can decrease to 20 per cent [36,37]. In contrast, DI resulted in spine unloading with a ~7% increase in ADC, indicating increased water content, as observed in the current study. This increase in water content was associated with a ~10% increase in the IVD protein content. While a portion of the protein and water increase may result from extracellular changes, it could also result from structural modifications of proteoglycans [38], or the synthesis of new proteoglycans within the IVD. With the current data, it is still unclear whether increased IVD protein is the cause or the result of changes in IVD water content.

The observed increase in IVD water and protein content could have implications for IVD nutrition. As discs are avascular, the exchange of metabolic substrates occurs via diffusion through the vertebral endplates. Therefore, change in IVD oncotic pressure may alter disc nutritional status. The current study was not designed to investigate disc nutrition; however, this does present a potential area of investigation for future studies, especially in the context of disc adaptations to long-term microgravity exposure.

Adaptations to spaceflight are often paralleled to aging on Earth [39]. Intervertebral aging is characterized by water loss and protein depletion [40,41], with a general decrease in collagen in the lumbar and cervical regions [41,42], and is associated with increased risk of IVD herniation [43,44]. In contrast to aging, the current study noted increased water and protein content with DI, suggesting the possibility of a different mechanism causing increased risk of disk herniation. However, it should be noted that the current study utilized a model to simulate microgravity exposure that may not completely reflect potential disc degeneration and the development of low back pain with missions to the International Space Station [24,45].

To date, there is a lack of data on cervical IVD changes with real or simulated spaceflight [3,46]. There are anatomical and biomechanical differences between the lumbar and cervical spine which could affect adaptations. In cervical intervertebral discs, the anulus fibrosus is not circular, but rather, crescent shaped, and serves as an interosseous ligament in the saddle joint between the vertebral bodies [47]. Additional study is needed with astronauts and microgravity exposure to determine whether these anatomical and biomechanical differences influence cervical IVD physiological and structural adaptations to exposure to microgravity.

### 4.1. Spaceflight Countermeasures

Spaceflight research has frequently involved the investigation of various countermeasures to prevent physiological deconditioning during flight. In a head-down bed rest model of microgravity exposure, exercise with lower body negative pressure has been shown to counteract lumbar spine deconditioning [48]. As thigh cuffs are often used by Russian cosmonauts to counteract the fluid shifts associated with exposure to microgravity, the current study investigated whether thigh cuffs would also assist in preventing spinal deconditioning; no differences in IVD properties were found with thigh cuff use.

Resistive exercise devices are used by astronauts during space [49], primarily to maintain muscle tone without affecting spine conditioning. With respect to spine health, additional exercises are needed to specifically target deep paravertebral muscles like *multifidus* [24]. The development of a mechanical countermeasure to avoid IVD degeneration would need to investigate different specific loading profiles to balance the influences of lower back pain, IVD load, and muscular stability [24,31,45]. Research from animal models suggests the existence of a dose-dependent relationship between loading and regenerative processes [50]. Low-volume loading at a low frequency appears to induce potentially regenerative mechanisms, including improvements in the disc proteoglycan content, matrix gene expression, rate of cell apoptosis, and improved fluid flow and solute transport [51]. Human studies are needed to confirm similar adaptations, investigating both lumbar and cervical regions [12].

### 4.2. Limitations

The current study utilized MRI measurements to determine changes in lumbar IVD water and protein content. It was believed that the resolution of the system used was sufficient to detect changes in these variables. While there may be some error in the absolute value of the measured variables due to the resolution limitations of the system used, directional changes were consistent for all study participants, supporting the conclusion that DI exposure results in increased IVD water and protein content.

The purpose of the current study was to determine IVD adaptations to DI exposure as potential mechanisms for IVD degeneration. While changes in disc water and protein content were observed, degeneration was not seen. This was expected, as participants were only exposed to DI for five days, which would not be enough to result in structural changes. Additional study would be needed to determine whether longer duration DI results in disc degeneration, and whether this is related to the observed acute changes in disc water and protein content.

To assess IVD adaptations, DI was used as a model of microgravity exposure. This model has been shown to mimic the fluid shifts and cardiovascular deconditioning commonly observed with spaceflight; however, it may not fully simulate changes in spine loading. With DI, normal respiration affects the buoyancy of the thorax, potentially placing oscillatory stress on the spine that is not present during spaceflight. As a model of microgravity exposure, the results from this study present potential avenues of investigation to determine whether similar adaptations occur during spaceflight, and how these adaptations may influence spine health with long-term microgravity exposure.

## 5. Methods

### 5.1. Subjects

Twenty healthy men were recruited for the study. Two individuals withdrew before testing for reasons unrelated to the dry immersion (DI) protocol (elbow injury and anxiety during the MRI exam). A total of eighteen subjects were included in the study and were randomly to assigned to the Control or Cuffs group (9/9 split). All subjects were informed of the experimental procedures and gave written consent before participating. The experimental protocol conformed to the standards set by the Declaration of Helsinki and was approved by the local Ethic Committee (CPP Est III: 2 October 2018, n° ID RCB 2018-A01470-55) and French Health Authorities (ANSM: 13 August 2018; ClinicalTrials.gov Identifier: NCT03915457). Baseline group characteristics are detailed in Table 3. There were no significant differences between the groups at baseline.

### 5.2. General Protocol

The study was conducted at the MEDES-IMPS space clinic, Toulouse, France. Participants arrived in the evening five days before DI (B-5) and left in the morning two days after DI (R+2). The experimental protocol included four days of ambulatory baseline measurements (B-4 to B-1), followed by five days (120 h) of DI (D1 to D5) and two days of ambulatory recovery (R0, R+1). Participants in the Cuffs group wore thigh cuffs during the five days of DI. The cuffs consisted of elastic straps that were adjusted for each participant to apply approximately 30 mmHg of pressure. Thigh cuffs were put on immediately prior to the onset of immersion, removed each evening (18:00 p.m.), and reapplied each morning (08:00 a.m.) of the DI period.

DI was conducted in accordance with previously described methodology [7]. With the exception of one participant in the Control condition and one in the Cuffs condition who completed the protocol individually, two participants, one Control and one Cuffs, underwent dry immersion simultaneously in the same room, in two separate baths. The thermoneutral water temperature was continuously maintained (32.5–33.5 °C). Ambient lighting was turned off from 23:00 p.m. to 07:00 a.m. each day. Daily hygiene, weighing, and some specific measurements required extraction from the bath. During these out-of-bath periods, subjects maintained a −6° head-down position. The total out-of-bath time for the 120 h of immersion was 9.7 ± 1.3 h. During DI, subjects remained immersed in a supine position for all activities, and were continuously observed by video monitoring. Body weight, blood pressure, heart rate, and body temperature were measured daily. Throughout the process, water intake was ad libitum and measured to ensure an intake of 35–60 mL/kg/day. The menu composition of each experiment day was identical for all participants, and the dietary intake was individually tailored and controlled during the study.

### 5.3. Magnetic Resonance Imaging

Magnetic resonance imaging (MRI) was conducted on B-3 and in the evening of D5. Subjects were maintained in a −6° head-down position between the DI bath and the MRI device in the hospital (Rangueil, Toulouse, France) to help preserve the DI effects. A 1.5T Siemens Magnetom Avanto syngo MR B17 was used for the MRI sequence acquisitions. MRI data were stored and analyzed at a later date using OleaSphere v3.0-SP16 (Olea Medical, La Ciotat, France) for the calculation of the apparent diffusion coefficient (ADC) map, T1 and T2 maps, relaxometry and distance measurements. To prevent interobserver variability, a single researcher performed all offline MRI measurements. Measurements pre and post were made in triplicate (standard deviation of 3%) with the mean value used for analysis. Technical issues prevented the collection of MRI data for one subject (Control condition); therefore, the data for 17 subjects were analyzed. 

#### 5.3.1. Disc Water Content

Diffusion-weighted imaging (DWI) is a specialized MRI technique that is particularly sensitive to random or “Brownian” microscopic motion. The displacement of diffusing water protons that occurs within a certain observation interval (diffusion time) can be quantified by a value called the apparent diffusion coefficient (ADC). ADC measurements are currently used in IVD research [15,52,53]. The ADC is computed by linear regression of the isotropic images (Iiso) from each diffusion weighting factor (b0-b800), according to the following formula: ADC = D = −1/b × ln (Iiso/I_0_).

The IVD water content was measured by sagittal ADC (ADC value in ms × 10^−3^ mm^2^/s). Sequences (FoV 250 mm, TR 2700 ms, TE 73 ms, with 1268 Hz/Px) were centered in six lumbar discs from T_12_ to S_1_, and were presented in sagittal view with the same anatomic landmarks. Measurements were focused in the IVD with a manual selection respecting the anatomical IVD signal and crosschecking the transversal plan to ensure precise anatomical measurement. The ROI in the NP was the same size in pre- vs. post- analysis, and was placed in the middle of the disc to get a good comparison between sequences. 

#### 5.3.2. Protein Content in IVD

The protein (glycosaminoglycan) content was measured on MRI T1 mapping sequences. The protein IVD content has been previously calculated by MRI [54,55,56], but to our knowledge, it has never been measured in a microgravity simulation study. A relaxometry analysis was performed with a T1 vibe sagittal multi echo (flips angles 2–7–10–15 degrees) with FoV 260 mm, TR 5.07 ms, TE 1.69 ms, and slice 5.0 mm. The region of interest (ROI) was the nucleus pulposus (NP) with the same anatomical landmarks of the IVD, and was centered in the sagittal plan. The selection cross checked pre/post sequences in order to get reliable measurements inside the NP. T1 maps were computed for both pre- and post- dry immersion exams, and were compared side by side. ROIs were drawn into the NP on the first exam and propagated to the second one (Figure 4).

#### 5.3.3. MRI Spinal Height and Lumbar Curvature

The spine height was measured as the distance between the occipital bone and the first sacral vertebra (S_1_) with the same spinal alignment, crosschecking the axial and transversal plans pre- vs. post-DI, as previously described [10]. The lumbar lordosis was measured as the vertebral wedging angle between L_1_ and the upper part of S_1_ in the median plane [24,57]. The MRI sequence characteristics were T2 sagittal images with TR 2650 ms, FoV 450 mm, TE 73 ms, and slice 5.0 mm.

#### 5.3.4. Plasma Volume Variations

Percent change in plasma volume (DPV) on D1-evening, D3-morning, D5-morning, D5-evening, and R0-morning vs. baseline (D_1_-morning before the onset of immersion) was estimated using hemoglobin and hematocrit counts (Dill and Costill formula).

### 5.4. Statistics

Data are presented as mean ± SD unless otherwise specified. A two-way repeated analysis of variance (ANOVA) was used to compare the pre/post measurements for the two groups. An adjusted *p* value of <0.05 was considered significant. In the case of no significant differences between the groups, the effects of DI for all of the participants were analyzed using paired t-tests. All of the analyses were performed using Prism GraphPad v8.3.0 software.

## Figures and Tables

**Figure 1 ijms-21-03748-f001:**
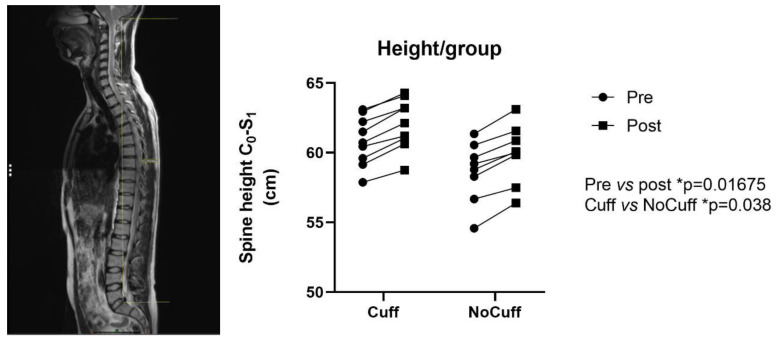
Magnetic resonance imaging (MRI) T2 sagittal sequence centered in the median plane. Spine height (distance occipital bone C_0_ to first sacral vertebra S_1_ as shown in yellow line) increased for each of subject, with a mean change from 59.81 to 61.06 cm (95% confidence interval (CI) of difference 1.428 to 1.065 cm; * *p* < 0.05).

**Figure 2 ijms-21-03748-f002:**
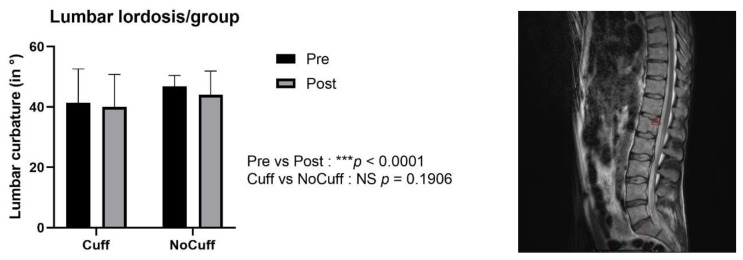
Magnetic resonance imaging (MRI) T2 sagittal sequence centered in the median plane. Lumbar lordosis (vertebral endplates L_1_-S_1_ in red line) decreased pre- vs post (*t*-test *** *p* < 0.0001; 95% CI from 39.50 to 48.45° in pre to 37.05 to 46.80° in post) with no effect of the countermeasure (*p* < 0.1906).

**Figure 3 ijms-21-03748-f003:**
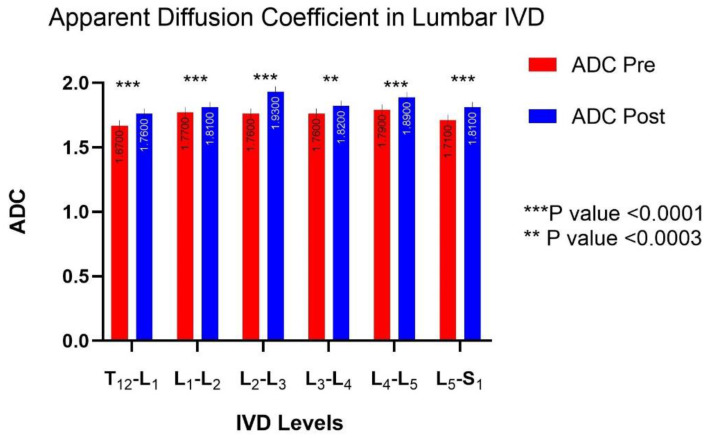
Apparent diffusion coefficient (ADC; value in ms ×10^−3^ mm^2^/s) per intervertebral lumbar disc level (** *p* < 0.001; *** *p* < 0.0001). The ADC increased significantly after dry immersion, with an average increase of 7.34 ± 2.23% (*p* < 0.05).

**Figure 4 ijms-21-03748-f004:**
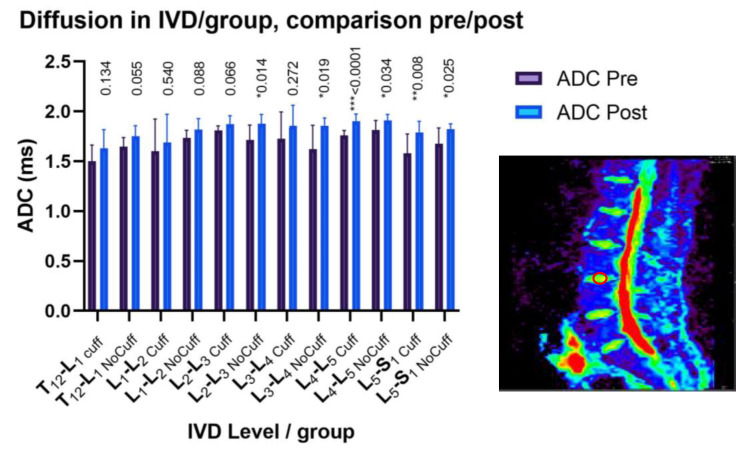
Apparent diffusion coefficient (ADC) in lumbar intervertebral discs (IVD) from the MRI sagittal view (region of interest in red circle inside IVD L3-L4). All of the lumbar levels significantly increased in water content after dry immersion (* *p* < 0.05; ** *p <* 0.01; *** *p* < 0.001). There were no statistical differences between the Cuffs and Control groups (*p* = 0.92).

**Figure 5 ijms-21-03748-f005:**
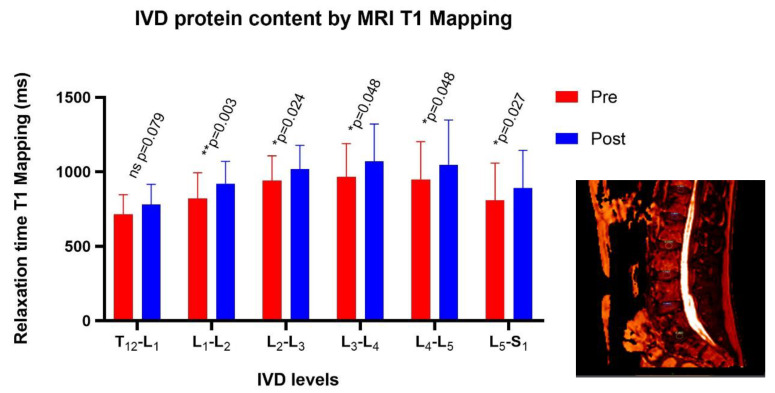
T1 mapping images pre vs. post are presented with an OleaSphere filter with the region of interest (ROI) inside the nucleus pulposus (NP). Protein content in the NP significantly increased after DI (* *p* < 0.05 and ** *p* < 0.01).

**Figure 6 ijms-21-03748-f006:**
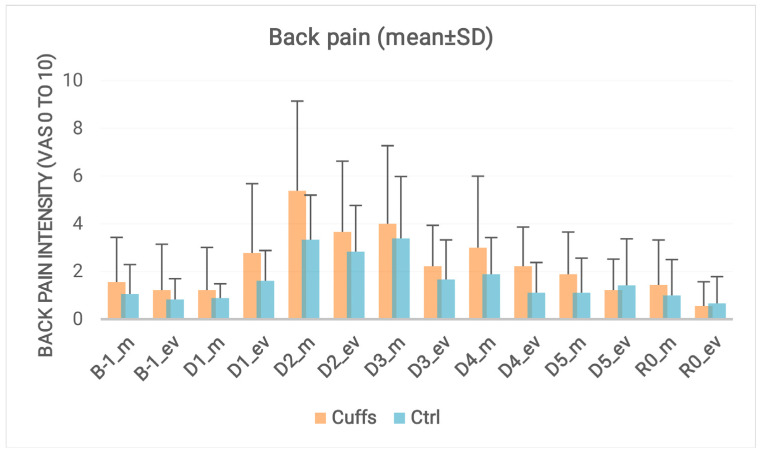
Back pain intensity (0 to 10 on a visual analog scale). Data collection before (B-1) to recovery (R0) every day for five days of dry immersion (D_1_ to D_5_) in the morning (m) and in the evening (ev). Two-way analysis of variance (ANOVA) revealed a significant global effect of DI (*p* < 0.0001), and no effect of countermeasure (*p* = 0.34).

**Table 1 ijms-21-03748-t001:** Plasma volume changes as a percentage of baseline (mean ± SD). * *p* < 0.05.

	D1_Evening	D3_Morning	D5_Morning	D5_Evening	R0_Morning
**Control**	−8.9 ± 4.6 *	−22.1 ± 6 *	−19.2 ± 8.5 *	−11.8 ± 6.4 *	−15.3 ± 6.7 *
**Cuffs**	−6.8 ± 5 *	−19.1 ± 5.4 *	−14.6 ± 4.3 *	−6.8 ± 5 *	−10 ± 4.4 *

**Table 2 ijms-21-03748-t002:** Blood Na+, K+, proteins, and osmolality. Values are mean ± SD; * *p* ≤ 0.05 vs. baseline; # *p* < 0.05 vs. Control; m—morning; e—evening.

	Control				Cuffs			
Variable	B-1_m	DI-1_e	DI-3_m	DI-5_m	B-1_m	DI-1_e	DI-3_m	DI-5_m
Sodium, mmol/L	141 ± 2	140 ± 2	139 ± 2	139 ± 2	141 ± 2	139 ± 3	138 ± 4 *	138 ± 5 *
Potassium, mmol/L	3.8 ± 0.2	4 ± 0.3 *	3.9 ± 0.2	3.9 ± 0.2	3.9 ± 0.2	4.0 ± 0.2	4.0 ± 0.2	4.0 ± 0.2
Proteins, g/L	67 ± 4	68 ± 2	73 ± 3 *	71 ± 4 *	67 ± 3	66 ± 3	68 ± 4 #	65 ± 4 #
Osmolality, mOsmol/kg	292 ± 5	294 ± 3	293 ± 4	293 ± 5	293 ± 4	294 ± 7	293 ± 5	292 ± 6

**Table 3 ijms-21-03748-t003:** Baseline group characteristics at B-2, mean ± standard deviation (SD). An unpaired T-test did not reveal any significant differences between groups.

	Age (y)	Height (cm)	Spine Height by MRI	Lumbar lordosis (°)	Weight (kg)	BMI (kg/m^2^)	VO_2_max (ml/min/kg)	HR (bpm)	T (°C)	SBP (mmHg)	DBP (mmHg)
Control (n = 9)	33.9 ± 7.1	176 ± 6	59.28 ± 0.37	46.82 ± 3.66	73.9 ± 7.5	23.9 ± 1.7	46.5 ± 8.1	57 ± 6	36.4 ± 0.3	115 ± 11	68 ± 5
Cuffs (n = 9)	34.1 ± 3.7	180 ± 4	61.45 ± 0.40	41.45 ± 11.17	74.3 ± 8.8	22.7 ± 1.8	46.9 ± 5.8	58 ± 8	36.4 ± 0.5	117 ± 10	68 ± 9

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
