# Peer review of "DI-5-Cuffs: Lumbar Intervertebral Disc Proteoglycan and Water Content Changes in Humans after Five Days of Dry Immersion to Simulate Microgravity"

_ijms, 2020, doi:10.3390/ijms21113748_

Round 1
Reviewer 1 Report
Brief summary
In their prospective study with 9/9 subjects the authors investigate IVD changes in response to simulated microgravity. They analyzed IVD protein and fluid content with MRI diffusion analysis. Additionally, the investigated the effects of thigh cuffs on IVD properties. They found an ADC increase inside the IVD.
Broad / specific comments
It is a well-written analysis. The study group is well defined and the study protocol described.
A grading for disc degeneration (influencing the water and GAG content) level in the measured IVDs is missing. Given ADC values argue for DD with Pfirrman Grade 2 to 3.
Measurement were taken in the NP with 1.5T. The discussion and the whole analysis is discussing on disc as a whole. The references dealt with 3.0T and gagCEST. Therefore, a validation is needed for the used technique in this study (do we measure what we want to measure?) – including an (measurement) error analysis. No inter- and intraobserver reliability is provided.
Whole height of the spine is given, but no pre vs post analysis of the changed curvature of the spine (for instance: less lordosis can increase the spine length without change of disc due to swelling)
Table 4: the Variances of ADC pre and ADC post are not provided No explanation is given for the different “increase” of ADC values.
Figure 2 is not readable. Where do the variances end?
Selected image of ADC: where is the ROI measured? Use appropriate image. It is difficult to determine the nucleus solely and a measurement error may be inevitable(voxel size, disc degeneration,…). How many times did you repeat the measurements, how high is the systematic error?
Discussion section:
Spine lengthening in not discussed in the context of spinal curvature.
The observation increase of the IVD volume is not discussed in the context of inactivated muscle and ligaments.
It would be interesting to now something about the possible time frame of loading and unloading and their effect on IVD “swelling”. Also, the osmotic nature of disc nutrition is unfortunately not discussed.
This study was conducted on lumbar spine. A direct correlation to cervical spine could not be seen by the reviewer.
With this study only the hyperhydration was proved, but not IVD degeneration. Please, be more precied in your argumentation.
References:
This frequency of self-citations should be avoided
Recommendation
I congratulate the authors to their study and all their efforts. Even tough, the question of the microgravity and their effects on intervertebral disc concerns only a small subset of our patients it is a new aspect, which should be taken into account.
Nevertheless, there are some major flaws. In my perception, the results in the presented form are not imperatively constructive for every reader. But, I am sure that the presented manuscript will benefit from a substantial revision. Additionally, a precise and detailed limitation section will attach much value to the manuscript
Author Response
We thank the reveiwer for all the comments.
Reviewer #1 Comments:
Brief summary
In their prospective study with 9/9 subjects the authors investigate IVD changes in response to simulated microgravity. They analyzed IVD protein and fluid content with MRI diffusion analysis. Additionally, the investigated the effects of thigh cuffs on IVD properties. They found an ADC increase inside the IVD.
Broad / specific comments
It is a well-written analysis. The study group is well defined and the study protocol described.
A grading for disc degeneration (influencing the water and GAG content) level in the measured IVDs is missing. Given ADC values argue for DD with Pfirrman Grade 2 to 3.
Response: Assessments of disc degeneration were not included in the study as we believed 5 days of DI would not be sufficient to induce disc degeneration. Pfirrman Grade was not changed after DI and a note of this has been added to the text.
Measurement were taken in the NP with 1.5T. The discussion and the whole analysis is discussing on disc as a whole. The references dealt with 3.0T and gagCEST. Therefore, a validation is needed for the used technique in this study (do we measure what we want to measure?) – including an (measurement) error analysis. No inter- and intraobserver reliability is provided.
Response: The use of the 1.5T MRI for the measures made in the current study has been validated in clinical practice and had been used in previous investigations of ACD (Spine. 2006; 31(14): 1547-1554). Interobserver variability was not a factor in the current study as single researcher performed all measurements. This has been added to the methods section.
Whole height of the spine is given, but no pre vs post analysis of the changed curvature of the spine (for instance: less lordosis can increase the spine length without change of disc due to swelling)
Response: Changes in lumbar lordosis have now been added to the paper in a new Figure 2 and in the baseline table. There was a significant reduction in curvature with DI for all participants with no effect of the thigh cuff countermeasure.
Table 4: the Variances of ADC pre and ADC post are not provided No explanation is given for the different “increase” of ADC values.
Response:
The L4-L5 and L5-S1 are more affected because of the specific pelvis position in retroversion with hip in flexion during the DI. That could increase the majored loss of lordosis in the low part of the lumbar spine. Moreover these 2 discs are the biggest IVD which could "receive" more water inside.
Figure 2 is not readable. Where do the variances end?
Response: We have deleted the table 3 in a figure 3 to be more readable and we have included the ADC mean values inside the graph.
Selected image of ADC: where is the ROI measured? Use appropriate image. It is difficult to determine the nucleus solely and a measurement error may be inevitable (voxel size, disc degeneration,…). How many times did you repeat the measurements, how high is the systematic error?
Response: An arrow has been added to the image to highlight the ROI for the ADC and T1-Mapping measurements. Both MRI measurement were conducted triplicate for the pre and post sessions with the average of the three used for analysis. Measurements had a standard deviation of 3%. This information has been added to the Methods section of the paper. The ROI in the NP was the same size in Pre vs Post analysis and placed in the middle of the disc in order to get a good comparison of the sequences. This information is added in the methods.
Discussion section
Spine lengthening is not discussed in the context of spinal curvature.
Response: Spine length is influenced by spine curvature and IVD thickness. Although lumbar curvature was decreased after DI, this change does not fully explain the observed increase in spine length. A sentence has been added to the discussion to indicate that the observed increase in spine length was also due to the reduction in curvature.
The observation increase of the IVD volume is not discussed in the context of inactivated muscle and ligaments.
Response: We have added a reference of Bailey et al. with data on ISS who suggest the correlation with paravertebral muscle atrophy, targeting the multifidus muscles.
It would be interesting to now something about the possible time frame of loading and unloading and their effect on IVD “swelling”. Also, the osmotic nature of disc nutrition is unfortunately not discussed.
Response: The time frame required for IVD “swelling” changes with DI or microgravity exposure would be an interesting area of investigation, but was outside the scope of the current study. As mentioned in the Discussion, similar changes in spine length were noted after only 3 days of DI exposure, but it is unclear how quickly this adaptation occurs.
The issue of disc nutritional changes with DI or microgravity exposure is an interesting area of investigation. Although this is outside the scope of the current study, the potential for changes in disc nutritional status has been added to the discussion.
This study was conducted on lumbar spine. A direct correlation to cervical spine could not be seen by the reviewer.
Response: We agree that the observed changes in the lumbar spine may not be directly correlated to changes in the cervical spine. However, Astronauts retuning from long duration spaceflight also report increased cervical spine pain with the possibility of herniation in this region. Therefore, future study should investigate the cervical spine to determine adaptations of IVDs in this region to microgravity exposure. There is a lack of data on cervical part.
With this study only the hyperhydration was proved, but not IVD degeneration. Please, be more precise in your argumentation.
Response: We agree that the current study only showed increased IVD water and protein content and not degeneration. As the study only used 5 days of DI exposure, it was highly unlikely that participants would exhibit evidence of disc degeneration. However, disc degeneration has been shown after long duration spaceflight. Therefore, the purpose of the current study was to determine potential mechanisms of degeneration by examining IVD short term adaptations with simulated microgravity exposure.
References
This frequency of self-citations should be avoided
Recommendation
I congratulate the authors to their study and all their efforts. Even tough, the question of the microgravity and their effects on intervertebral disc concerns only a small subset of our patients it is a new aspect, which should be taken into account.
Nevertheless, there are some major flaws. In my perception, the results in the presented form are not imperatively constructive for every reader. But, I am sure that the presented manuscript will benefit from a substantial revision. Additionally, a precise and detailed limitation section will attach much value to the manuscript
Response: Alterations have been made to the presentation of the paper. In addition, a limitations section has now been added to the discussion.
Reviewer 2 Report
The authors present a fundamental study on the effects of microgravity onto the protein and fluid content of IVD. Their model comprises dry immersion rather than exposure to real zero gravity as in space or microgravity as obtained in random positioning machines (RPM) on the earth.
The authors present a carefully prepared manuscript on the influence of simulated microgravity using Dry Immersion. I would like to congratulate to a fascinating study, analysing carefully the subjects that volunteered using MRI and latest imaging standards.
Material and Methods should go at the end of this journal format. Thus, results should become section 3 then followed by the discussion, and Material and Methods should be listed last.
Introduction:
It is uncertain whether herniations are the main cause of low back pain (LBP) for astronauts. The problem arises merely from the decompression, which causes swelling overall and not necessarily herniations. Accelerated disc degeneration by fluctuations in fluid contents and swelling and compression should also be added as an important factor for the development of LBP.
Two important studies are to be mentioned here, how prolonged unloading to discs can lead to disc degeneration. These articles should be either added to the introduction or discussed their results.
1. Bailey JF, et al. (2018) From the international space station to the clinic: how prolonged unloading may disrupt lumbar spine stability. Spine J 18(1):7-14 https://doi.org//10.1016/j.spinee.2017.08.261
2. Bailey JF, et al. (2014) Effect of microgravity on the biomechanical properties of lumbar and caudal intervertebral discs in mice. J Biomech 47(12):2983-2988 https://doi.org//10.1016/j.jbiomech.2014.07.005
Experimental design
The ethics is important and has been given in full detail for this study.
The current study does not present any considerations on changed biomechanics after prolonged exposure to DI of the subjects. Maybe this should be discussed.
Author Response
Reviewer 2 Comments:
The authors present a fundamental study on the effects of microgravity onto the protein and fluid content of IVD. Their model comprises dry immersion rather than exposure to real zero gravity as in space or microgravity as obtained in random positioning machines (RPM) on the earth.
The authors present a carefully prepared manuscript on the influence of simulated microgravity using Dry Immersion. I would like to congratulate to a fascinating study, analysing carefully the subjects that volunteered using MRI and latest imaging standards.
Material and Methods should go at the end of this journal format. Thus, results should become section 3 then followed by the discussion, and Material and Methods should be listed last.
Response: The format of the paper has been changed so that Materials and Methods are now the last section.
Introduction:
It is uncertain whether herniations are the main cause of low back pain (LBP) for astronauts. The problem arises merely from the decompression, which causes swelling overall and not necessarily herniations. Accelerated disc degeneration by fluctuations in fluid contents and swelling and compression should also be added as an important factor for the development of LBP.
Response: The cause of LBP after spaceflight is likely due to different mechanism than the LBP that participants experienced in the current study. With only 5 days of DI, IVD degradation is unlikely. Following DI, all participants recovered very quickly, even if they experienced LBP during DI. In contrast, Astronauts report LBP for a much longer time after long duration spaceflight which could be related to LVD degeneration and the compressive load of gravity.
Two important studies are to be mentioned here, how prolonged unloading to discs can lead to disc degeneration. These articles should be either added to the introduction or discussed their results.
- Bailey JF, et al. (2018) From the international space station to the clinic: how prolonged unloading may disrupt lumbar spine stability. Spine J 18(1):7-14 https://doi.org//10.1016/j.spinee.2017.08.261
- Bailey JF, et al. (2014) Effect of microgravity on the biomechanical properties of lumbar and caudal intervertebral discs in mice. J Biomech 47(12):2983-2988 https://doi.org//10.1016/j.jbiomech.2014.07.005
Response: As requested, we have included these two important references in our introduction and discussion. (ref 24 and 45).
Experimental design
The ethics is important and has been given in full detail for this study.
The current study does not present any considerations on changed biomechanics after prolonged exposure to DI of the subjects. Maybe this should be discussed.
Response: The objective of the current study was to determine changes in IVD water and protein content after 5 days of DI. Changes in spine biomechanics were not extensively discussed in this paper as this was not a primary objective and were addressed in our previous study involving 3 days of DI exposure (Pain Res Manag. 2017;2017:9602131).